# Comparison of BCG Tokyo172 Strain Induction Therapy Between Low Dose and Standard Dose for Non-Muscle Invasive Bladder Cancer: Intravesical Instillation of BCG Tokyo172 Strain

**DOI:** 10.3390/biomedicines13010174

**Published:** 2025-01-13

**Authors:** Hideyuki Isobe, Fumitaka Shimizu, Takeshi Ieda, So Nakamura, Naoko Takazawa, Hanna Suetsugu, Kazunori Kajino, Shu Hirai, Hisashi Hirano, Katsuhito Yuzawa, Shigeo Horie

**Affiliations:** 1Department of Urology, Juntendo Tokyo Koto Geriatric Medical Center, Tokyo 136-0075, Japan; so-nakamura@juntendo.ac.jp (S.N.); matsu@juntendo.ac.jp (N.T.); hsuetsugu0829@gmail.com (H.S.); 2Department of Urology, Juntendo University Graduate School of Medicine, Tokyo 113-8431, Japan; f_simizu@juntendo.ac.jp (F.S.); tieda@juntendo.ac.jp (T.I.); k.yuzawa.sc@juntendo.ac.jp (K.Y.); shorie@juntendo.ac.jp (S.H.); 3Department of Pathology, Juntendo Tokyo Koto Geriatric Medical Center, Tokyo 136-0075, Japan; kajino@juntendo.ac.jp (K.K.); shuh@juntendo.ac.jp (S.H.); 4Department of Urology, Otakanomori Hospital, Chiba 277-0863, Japan; hhirano@juntendo.ac.jp

**Keywords:** non-muscle invasive bladder cancer, low-dose BCG, standard-dose BCG, intravesical induction therapy

## Abstract

Objectives: The aim of this study was to identify factors that predict recurrence by comparing low-dose and standard-dose Bacillus Calmette-Guérin (BCG) induction therapy in patients with non-muscle invasive bladder cancer (NMIBC). Methods: A total of 273 consecutive NMIBC patients who received low-dose (40 mg) or standard-dose (80 mg) BCG intravesical instillation therapy between January 2004 and December 2023 were analyzed. Recurrence-free survival (RFS) rates were assessed using the Kaplan–Meier method with the log-rank test. Univariate and multivariate Cox proportional hazards regression analyses were used to identify independent predictors of recurrence based on the Club Urológico Español de Tratamiento Oncológico (CUETO) criteria. Results: The log-rank test showed that older age, low BCG dose, number of tumors, and a history of recurrence increased the risk of recurrence significantly. Regarding older patients, recurrence rates were similar between the two dose groups. However, younger patients had significantly lower recurrence rates with standard-dose BCG compared to low-dose BCG. Multiple Cox regression analysis confirmed that older age, low-dose BCG, greater than three tumors, and a history of recurrence were significant predictors of recurrence. Conclusions: In this study, we found that low-dose BCG induction therapy was associated with higher relapse rates compared with standard doses, especially in younger patients. Age-related differences in the immune response to BCG may influence these relapse patterns.

## 1. Introduction

NMIBC accounts for about 75% of bladder cancers and consists of Ta, which remains in the bladder mucosa, Tis, an intraepithelial carcinoma, and T1, which infiltrates submucosal tissue [1]. Many patients receive intravesical instillation therapy with chemotherapy or BCG to prevent recurrence after transurethral resection of the bladder tumor (TURBT) [2]. The primary goal in NMIBC is to prevent recurrence of cancer and progression to muscle invasive cancer. BCG intravesical instillation therapy is the most common adjuvant treatment for NMIBC. Its effectiveness and safety have been demonstrated in multiple randomized controlled trials (RCTs) and meta-analyses [3]. There are two methods of administering BCG: induction therapy, in which it is administered 6–8 times after TURBT, and maintenance therapy, in which BCG is administered continuously for 1–3 years afterwards. International guidelines recommend BCG intravesical instillation for patients with intermediate- and high-risk NMIBC to reduce the risk of recurrence and progression [4]. However, there are often cases where it is difficult to continue treatment due to side effects from intravesical BCG injections. In particular, BCG maintenance therapy has been problematic due to its low completion rate due to adverse events and is, therefore, controversial. Key questions remain unresolved regarding the optimal BCG dose, duration of administration, and strain selection [5]. The European Organization for Research and Treatment of Cancer (EORTC) established recurrence and progression scores based on seven clinical trials focusing on intravesical chemotherapy [6]. However, as intravesical BCG therapy has become the standard treatment, factors related to recurrence and progression have changed. Club Urológico Español de Tratamiento Oncológico (CUETO) created the CUETO scoring system based on data from four clinical trials that included intravesical BCG therapy (Table 1) [7]. This study compares low-dose and standard-dose BCG Tokyo172 strain induction therapy of NMIBC for preventing recurrence and investigates prognostic factors associated with recurrence using the CUETO criteria.

## 2. Materials and Methods

Data from patients with NMIBC treated with intravesical low-dose or standard-dose BCG therapy were retrospectively reviewed. This retrospective study adhered to the guidelines established by the Declaration of Helsinki and was approved by the Institutional Review Board of Juntendo University School of Medicine (approval no. E24-0242). Patients were treated between January 2004 and December 2023 in Juntendo Koto Geriatric Medical Center and Juntendo University Hospital. All the patients underwent complete TUR.

### 2.1. Inclusion Criteria

The eligibility criteria included (1) no prior BCG (Tokyo172) maintenance therapy, (2) no prior intravesical induction therapy other than BCG (Tokyo172), (3) no untreated upper urinary tract cancer (unless tumor invasion was suspected during treatment), and (4) follow-up evaluation at least once after BCG induction therapy.

### 2.2. Study Treatment

A total of 40 mg or 80 mg of BCG (Tokyo172) was dissolved in 40 mL of normal saline, instilled into the bladder, held for one hour, and then voided. The procedure was once a week for six weeks. An effectiveness evaluation was performed every 3 months after the completion of intravesical instillation therapy using cystoscopy and urine cytology. Recurrence was defined as the presence of a papillary tumor or irregular mucosa confirmed by cystoscopy after the first BCG, followed by surgery and confirmation of recurrence by histopathological examination. CT scans were performed every 6 months to 1 year to monitor for upper urinary tract recurrence. Side effects were evaluated using Common Terminology Criteria for Adverse Events (CTCAE) Version4.0.

### 2.3. Statistical Analysis

Recurrence-free survival (RFS) rates were calculated using the Kaplan–Meier method, with comparisons made using the log-rank test. The Club Urológico Español de Tratamiento Oncológico (CUETO) scoring system was used to assess recurrence and progression risk (Table 1) [7]. This system consists of factors such as sex, age, history of recurrence, number of tumors, T classification, tumor grade, and concurrent carcinoma in situ (CIS). Univariate and multivariate Cox proportional hazards regression analyses were conducted to identify independent factors predicting recurrence based on BCG dose and CUETO criteria. Patients without recurrence or progression were censored at their last follow-up date. A *p* value of ≤0.05 was considered statistically significant. JMP18.1.1 (SAS Institute Inc., Cary, NC, USA), was used for statistical analyses.

## 3. Results

### 3.1. Patient Characteristics

A total of 273 participants were included in this study. Table 2 summarizes the patient characteristics by BCG dose based on the CUETO criteria. A total of 151 patients received low-dose BCG, and 122 received standard-dose BCG. No significant differences were observed between the two groups in terms of sex, history of recurrence, number of tumors, or concurrent CIS. However, low-dose BCG was more frequently administered to elderly patients. The standard-dose group included a higher proportion of pTa cases and high-grade tumors, while the low-dose group had a greater proportion of pT1 cases. A total of 12.1% of patients (33/273) underwent a second TURBT, with 32 cases of pT1 and one case of pTa. The pathological findings of the second TURBT were pT0 in 19 cases, pTa in 3 cases, pT1 in 7 cases, and pTis in 4 cases.

### 3.2. Outcome of the Low-Dose Group

The completion rate for all six instillations in the low-dose group was 95.3% (144 out of 151), with no significant differences between patients younger than 70 years (95.3%) and those 70 years or older (95.0%). Upper urinary tract recurrence was observed in 2.1% (three cases). Adverse events were reported in 52.3% (79 out of 151) of patients, including 68 cases of Grade 1, 6 cases of Grade 2, 5 cases of Grade 3, and no cases of Grade 4 or 5. There were three deaths attributed to bladder cancer during follow-up (one patient younger than 70 years, two patients aged 70 years or older), all in the high- or very high-risk groups. One patient aged 70 years or older underwent total cystectomy.

### 3.3. Outcome of the Standard-Dose Group

In the standard-dose group, the completion rate for all six instillations was 100% (122 out of 122). Adverse events were reported in 52.5% (64 out of 122) of patients, including 63 cases of Grade 1, 1 case of Grade 2, and no cases of Grade 3, 4, or 5. There were two deaths attributed to bladder cancer during follow-up (one patient younger than 70 years, one patient aged 70 years or older), both in the high-risk group.

### 3.4. Survival Analyses

Kaplan–Meier curves comparing RFS by BCG dose, age, history of recurrence, and number of tumors are shown in Figure 1a–d. The log-rank test indicated a significantly higher recurrence rate in patients who received low-dose BCG, were older, had a history of recurrence, or had greater than three tumors. Kaplan–Meier curves stratified by age, history of recurrence, and number of tumors are shown in Figure 2a–f. Low-dose BCG induction therapy was associated with significantly lower RFS in those under 70 years of age, those with a history of recurrence, and those with three or less tumors.

Univariate and multivariate analyses using the Cox proportional hazards model identified the following factors as significant predictors of recurrence: older age, history of recurrence, greater than three tumors, and low-dose BCG therapy (Table 3).

## 4. Discussion

Intravesical BCG instillation therapy is recommended for intermediate- and high-risk NMIBC to reduce the risk of recurrence and progression of NMIBC [8,9]. Various approaches have been investigated to optimize its efficacy in preventing tumor recurrence and minimize the occurrence of side effects, but no definitive conclusions have been reached regarding the optimal dose, duration of administration, or the preferred BCG strain. In this study, we retrospectively analyzed prognostic factors for recurrence following induction therapy with low-dose and standard-dose BCG intravesical instillation.

Previous studies on low-dose intravesical BCG induction therapy have reported recurrence rates of 20 to 32% [10,11,12]. However, direct comparisons with our study are challenging due to variations in sample sizes and observation periods across the reports. In our study, the recurrence rate was relatively high in the low-dose group. This may be attributable to the high recurrence scores assigned by the CUETO risk classification for factors such as age (older than 70 years) and concurrent CIS. Among the 142 cases analyzed, 99 patients (69.7%) were older than 70 years, and 59 patients (41.5%) had concurrent CIS, all of which are relatively high proportions. There are conflicting reports regarding the therapeutic efficacy of low-dose BCG compared to standard-dose BCG. Irie et al. reported that low-dose BCG using the Tokyo 172 strain demonstrated a comparable therapeutic effect while significantly reducing the incidence of side effects compared to standard-dose BCG [12]. Oshima et al., in a post-marketing survey of BCG Tokyo 172 strains, reported no significant differences in efficacy rates between the standard-dose and low-dose BCG groups. Additionally, no significant differences were observed between the two groups in terms of Ta-T1 and intraepithelial carcinoma, histological atypia, tumor depth, or the number of tumors [13]. On the other hand, Yokomizo et al. conducted a prospective study comparing low-dose BCG with standard-dose BCG for the treatment of NMIBC and CIS. Although the incidence of side effects was significantly lower in the low-dose group, the response rate of the low-dose group could not be proven non-inferior to the standard-dose group [14]. Ostrowski et al. also reported, using propensity scores, that low-dose BCG was associated with a significantly higher risk of recurrence than standard-dose BCG [15]. In our study, no significant difference in adverse events (AEs) was observed between the standard-dose and low-dose groups. In a meta-analysis on RFS rate by BCG dose of several strains that were used including the Tokyo172 strain, Zeng et al. reported no significant difference in non-relapse and non-progression survival between the low-dose group (1/3–1/2 dose) and the standard-dose group (80–120 mg) and reported significantly lower systemic or serious adverse effects in the low-dose group (relative risk: RR 0.52, [0.36–0.74]; 95% confidence interval: CI) [15]. Quan et al. also reported that the standard-dose group (80–120 mg) was significantly more effective (hazard ratio: HR 1.17 [1.006–1.30]; 95% CI) than the low-dose group (1/3–1/2 dose), while no significant differences were observed in non-progression survival, cause-specific survival, or overall survival [16]. According to Matsumoto et al., for some high-risk groups (Grade3, pT1), the recurrence rate tended to be higher in the low-dose BCG group [17]. Shiozaki et al. reported that the response rate was significantly higher in the low-dose BCG intravesical instillation group than in the standard-dose group, but the response rate was not low, even in the high-dose group, and no clear significant differences in response rate or non-relapse rate were observed in the other stratified parameters [18]. Thus, the superiority of low-dose or standard-dose treatment for NMIBC remains controversial.

The multivariate analysis in this study identified low-dose BCG, older age, history of recurrence, and tumor size as independent risk factors for recurrence. Low-dose BCG was associated with higher recurrence rates, particularly in younger patients. Therefore, reducing the BCG dose may not be necessary for younger individuals. In older patients, the therapeutic effect of BCG may be diminished due to declining immune function. It is reported that aging is a multifaceted process, involving numerous molecular and cellular mechanisms in the context of different organ systems [19]. A crucial component of aging is a set of functional and structural alterations in the immune system that can manifest as a decreased ability to fight infection, diminished response to vaccination, increased incidence of cancer, and higher prevalence of autoimmunity and constitutive low-grade inflammation, among others. In the future, treatment strategies should be considered according to age and immune function in addition to the malignancy and type of tumor. Liu et al. also reported that young age and adequate BCG are key factors for optimal BCG treatment efficacy in NMIBC [20].

Efforts are underway overseas to develop more immuno-oncological treatments for BCG-resistant NMIBC [21,22,23]. According to the NCCN guidelines, pembrolizumab, an anti-PD-1 antibody inhibitor, is recommended for the treatment of BCG-unresponsive carcinoma in situ (CIS) and high-risk papillary tumors. Additionally, Nadofaragene firadenovec, a gene therapy based on an adenovirus vector, is approved for treating high-risk, BCG-resistant NMIBC patients with carcinoma in situ [24]. Several other promising investigational drugs have been identified. TAR-200 is a bladder-specific drug delivery system designed to gradually release gemcitabine within the bladder [25]. CG0070 is an oncolytic virus therapy incorporating the GM-CSF gene into an adenovirus, engineered to selectively replicate in tumor cells with RB gene mutations or deletions [26]. It is being evaluated as a monotherapy or in combination with pembrolizumab [27]. N-803, an interleukin-15 receptor agonist immunotherapy, has been designated as a breakthrough therapy by the FDA when used in combination with BCG [23]. Clinical trials have reported a complete response rate of 62% and a duration of response exceeding 47 months. Furthermore, TAR-210, an intravesical drug delivery system for administering erdafitinib, is being explored as a potential treatment for metastatic urothelial carcinoma and is currently undergoing a Phase I trial [28]. In Japan, there is a growing hope for introducing treatments tailored to age and immune response to prevent NMIBC recurrence.

This study has several limitations. First, although the present study did not incorporate maintenance therapy after BCG induction therapy, the evidence from RCT of BCG maintenance therapy suggests that a 3-year maintenance schedule of BCG intravesical instillation therapy (a total of 27 infusions within 3 years), first described by Lamm et al., further reduces recurrence in high-risk cases, and the data for the first year showed no significant difference in cancer progression and survival rate [29]. This was also not true for intermediate-risk cases [30]. Other RCTs of maintenance therapy have also reported that the effect of maintenance therapy is not important in terms of recurrence or progression [31,32]. The low completion rate of maintenance therapy has long been noted, and the results of future case series are awaited.

Second, the pathological diagnoses were performed by pathologists at each institution, and the results were pooled and retrospectively reviewed, so the results are not those of a single pathologist. We believe that future studies should be based on the diagnostic results of a single pathologist.

## 5. Conclusions

This study suggests that low-dose BCG may lead to higher recurrence rates, particularly in younger NMIBC patients. Age-related differences in immune response to BCG could play a role in this pattern. Reducing the BCG dose may not be necessary for younger patients.

## Figures and Tables

**Figure 1 biomedicines-13-00174-f001:**
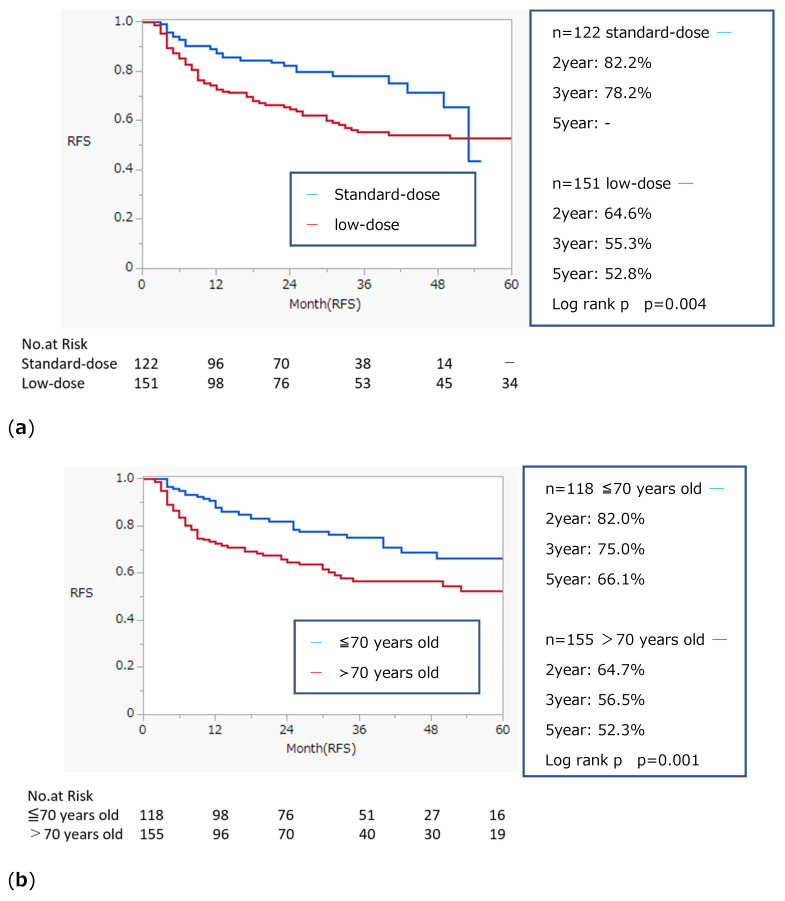
Kaplan–Meier curve and recurrence-free survival on BCG induction therapy for patient with NMIBC. (**a**) Comparison between standard dose and low dose; (**b**) Comparison between under and over 70 years old; (**c**) Comparison between primary and a history of recurrent disease; (**d**) Comparison between number of tumor 3 or less and greater than 3.

**Figure 2 biomedicines-13-00174-f002:**
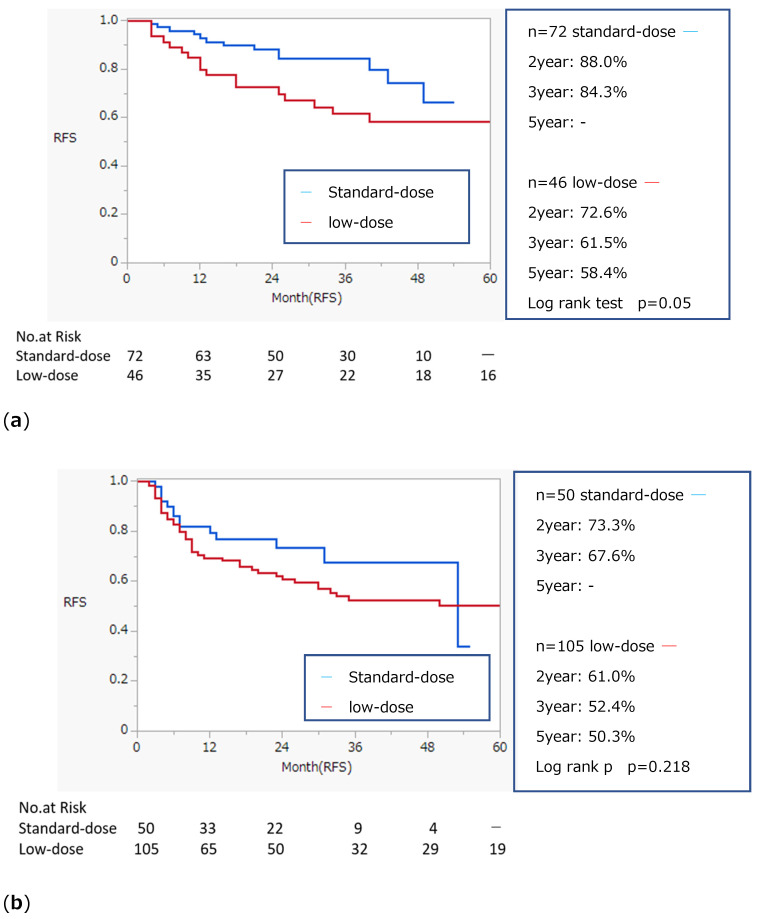
Stratified Kaplan–Meier curve and recurrence-free survival on BCG induction therapy for patient with NMIBC. (**a**) Comparison between low dose and standard dose under 70 years old; (**b**) Comparison between low dose and standard dose over 70 years old; (**c**) Comparison between low dose and standard dose on primary disease; (**d**) Comparison between low dose and standard dose on a history of recurrent disease; (**e**) Comparison between low dose and standard dose on 3 or less tumors; (**f**) Comparison between low dose and standard dose on greater than 3 tumors.

**Table 1 biomedicines-13-00174-t001:** CUETO risk table: Spanish Urological Oncology Group.

	Item	Recurrence Score	Progression Score
Sex	Male	0	0
	Female	3	0
Age	<60	0	0
	60~70	1	0
	>70	2	2
History of recurrence	Primary	0	0
	Recurrent	4	2
Number of tumors	<3	0	0
	≥3	2	1
T classification	Ta	0	0
	T1	0	2
Concurrent CIS	No	0	0
	Yes	2	1
Grade	G1	0	0
	G2	1	2
	G3	3	6
Total Score		0–16	0–14
**Risk Classification According to Applicable Items**			
		**Recurrence Risk**	**Progression Risk**
Low		0~4	0~4
Intermediate-low		5~6	5~6
Intermediate-High		7~9	7~9
High		10~16	10~14

**Table 2 biomedicines-13-00174-t002:** Patient characteristics (%).

		Low-Dose BCG n = 151	Standard-Dose BCG n = 122	*p* Value
**Sex**	male	122 (80.8)	95 (77.9)	0.553
	Female	29 (19.2)	27 (22.1)	
Age	≤70	46 (30.5)	72 (59.0)	<0.001
	>70	105 (69.5)	50 (41.0)	
History of recurrence	Primary	80 (53.0)	68 (55.7)	0.651
	Recurrent	71 (47.0)	54 (44.3)	
Number of Tumors	<3	122 (80.8)	109 (89.3)	0.052
	≥3	29 (19.2)	13 (10.7)	
T-classification	pTis	27 (17.9)	24 (19.7)	0.707
	pTa	51 (33.8)	69 (56.6)	<0.001
	pT1	73 (48.3)	29 (23.8)	<0.001
Concurrent CIS	Yes	78 (51.7)	51 (41.8)	0.106
	No	73 (48.3)	71 (58.2)	
G-classification	Grade 1	51 (33.8)	8 (6.6)	<0.001
	Grade 2–3	100 (66.2)	114 (93.4)	

**Table 3 biomedicines-13-00174-t003:** Univariate and multivariate analyses according to the CUETO classification in the RFS of 273 cases.

Item	Univariate			Multivariate		
	Hazard Ratio	95% CI	*p*	Hazard Ratio	95% CI	*p*
Female (vs. Male)	1.04	(0.61–1.69)	0.8751	—	—	—
Age > 70 (vs. ≤70)	1.82	(1.19–2.85)	0.0053	1.63	(1.05–2.62)	0.0336
Recurrent (vs. Primary)	1.66	(1.10–2.53)	0.0157	1.79	(1.13–2.85)	0.0126
Greater than 3 Tumors (vs. 3 or less)	1.65	(0.99–2.63)	0.0441	1.83	(1.09–2.94)	0.0157
T1 (vs. Ta)	1.23	(0.80–1.86)	0.3371	—	—	—
G2-3 (vs. G1)	0.84	(0.54–1.37)	0.4768	—	—	—
Concurrent CIS Yes (vs. No)	0.82	(0.54–1.24)	0.3508	—	—	—
BCG low-dose (vs. standard-dose)	1.91	(1.23–3.07)	0.0037	1.76	(1.05–3.01)	0.0358

## Data Availability

The data presented in this study are available on request from the corresponding author due to privacy and ethical restrictions.

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
