# Peer review of "Comparison of BCG Tokyo172 Strain Induction Therapy Between Low Dose and Standard Dose for Non-Muscle Invasive Bladder Cancer: Intravesical Instillation of BCG Tokyo172 Strain"

_biomedicines, 2025, doi:10.3390/biomedicines13010174_

Round 1
Reviewer 1 Report
Comments and Suggestions for Authors
It is stated that ‘All the patients underwent complete TUR’. Is it correct to understand that a diagnosis of Ta/T1 was obtained in all cases after the muscle layer sampling was confirmed by pathological evaluation? Also, please tell us the percentage of cases that underwent a second TUR and cases that were completed after the first TUR.
This study confirmed the efficacy of BCG "induction" therapy at two doses, but it would have been easier to understand if this point had been clearly stated in the title. Looking only at the title, it is unclear that not all cases in this study received maintenance therapy.
Please divide the results into sub-sections. It is difficult to read.
2.1% of patients had upper urinary tract recurrence, but the upper urinary tract examination details are not stated in the Methods section. Please indicate the timing of the examination and the detailed method.
The way the Kaplan-Meier curve is described is problematic, so please revise it in accordance with standard statistical methods.
Please explain why you did not conduct a study using propensity score matching. It would usually be considered appropriate to do so.
You used the CUETO classification in this study, but I think the EAU classification places more importance on the number of tumors (single or multiple). In clinical practice, the clinical course differs between single and multiple tumors. Please explain the rationale for using the CUETO classification. Please tell us whether the research results would change if this information were added.
Reviewer 2 Report
Comments and Suggestions for Authors
Considering the meaning of the research, he considers that the work is timely, because it is also a negative work, which shows us that there really are problems with minimizing the dose with BCG, something that is highly valued.
Methodologically I think it has been explained correctly, they explain the methodology, except that something very important is not clear to me, who and why? decided that one group would be given the standard dose and the other half the dose. It is too important ethically because we would be talking about a clinical trial and not just a retrospective analysis.
Round 2
Reviewer 1 Report
Comments and Suggestions for Authors The abbreviation for Common Terminology Criteria for Adverse Events is "CTCAE." The words sex and gender have entirely different meanings. This study probably should use the term "sex." Please seek guidance from a statistician on how to describe the Kaplan-Meier curve. Unfortunately, your description is not suitable for a scientific paper. Could you also tell us what software you used for the statistical processing? The acknowledgments section should explain the people who contributed to the research other than the co-authors and the research funding used.
Reviewer 2 Report
Comments and Suggestions for Authors
I thank the author for clarifying my doubt. I consider that the work is suitable for publication.
Author Response
→Thank you very much for reviewing my manuscript.
Round 3
Reviewer 1 Report
Comments and Suggestions for Authors
Does this mean that there were no censored cases in your Kaplan-Meier curve? Does this mean that all cases cleared the observation period without loss and only events (in this case, intravesical tumor recurrence) were observed? There is no table of numbers at risk; to me, at least, it does not look like a standard Kaplan-Meier curve. I think there is insufficient information, but what do you think?
Dr. Takeshi Ieda, Dr. Fumitaka Shimizu, Dr. Hisashi Hirano, and Dr. Shigeo Horie are co-authors of your paper. As I pointed out last time, there is no need to list co-authors in the acknowledgments section.
